# A Method for Extracting Debye Parameters as a Tool for Monitoring Watered and Contaminated Soils

**DOI:** 10.3390/s22207805

**Published:** 2022-10-14

**Authors:** Andrea Cataldo, Iman Farhat, Lourdes Farrugia, Raffaele Persico, Raissa Schiavoni

**Affiliations:** 1Department of Engineering for Innovation, University of Salento, 73100 Lecce, Italy; 2Department of Physics, University of Malta, 2080 Msida, Malta; 3Department of Environmental Engineering DIAM, University of Calabria, 87036 Rende, Italy

**Keywords:** microwave reflectometry, FDR measurements, bifilar probe, soil pollution, dielectric permittivity, Debye law

## Abstract

Soil monitoring is a key topic from several perspectives, such as moisture level control for irrigation management and anti-contamination purposes. Monitoring the latter is becoming even more important due to increasing environmental pollution. As a direct consequence, there is a strong demand for innovative monitoring systems that are low cost, provide for quasi-real time and in situ monitoring, high sensitivity, and adequate accuracy. Starting from these considerations, this paper addresses the implementation of a microwave reflectometry based-system utilizing a customized bifilar probe and a miniaturized Vector Network Analyzer (m-VNA). The main objective is to relate frequency-domain (FD) measurements to the features of interest, such as the water content and/or the percentage of some polluting substances, through an innovative automatable procedure to retrieve the Debye dielectric parameters of the soil under different conditions. The results from this study confirm the potential of microwave reflectometry for moisture monitoring and contamination detection.

## 1. Introduction

The monitoring of soil moisture content [1,2] and plant water status is of particular importance since these conditions are essential for proper management of nutrition and irrigation [3]. Proper monitoring would allow for efficient irrigation management that saves water, energy, fertilizer use, and time [4], improving the quantity and quality of production. Sustainable soil preservation is relevant for monitoring watered soils and safeguarding food production, as well as for the preservation of their associated ecosystem services, that are increasingly threatened by pollution. Nowadays, the problem of environmental pollution [5] is growing, especially in soil. The latter absorbs a wide variety of harmful substances, from heavy metals to organic pollutants and microplastics, and some unsustainable agricultural practices continue to contaminate the soil, mostly due to the use of copper and cadmium-based fertilizers [6] or pesticides [7]. In addition, agricultural practices are not the only sources of soil pollution: poorly managed waste and industrial activities [8] are also responsible for soil contamination. Another source of concern for soil contamination is oil leakage that flows out of broken oil-pipelines [9]. Oil leakages may spread over large areas, causing irreparable damage without being detected. As a result, the presence of these soil pollutants causes a chain reaction, altering soils’ biodiversity, nutrients, groundwater, and reducing soil organic matter.

However, the wide variety of contaminants, soils, and climatic conditions results in high costs for monitoring and comprehensive assessment of soil quality and land pollution. For all these reasons, there is a great need to monitor soils for different purposes, from pollution prevention to water content control, simultaneously ensuring high reliability and low-cost methods.

## 2. State-of-the-Art Methods and the Proposed Solution

Existing controls usually rely on highly sophisticated [10] and expensive methods (i.e., the neutron scattering method [11] or the gamma ray attenuation method [12]), while other techniques are highly invasive and require repeated drilling to collect samples for physical and chemical analysis [13]. In addition, drilling techniques can change the concentration of harmful substances in the soil and are inadequate to monitor large areas of land. Furthermore, technologies based on dielectric spectroscopy using resonant cavities and/or transmission lines [14,15] necessitate extensive sample machining and, in the case of resonant cavities, measurements can only be conducted at a single frequency, whereas multi-frequency measurements are ideal for retrieving physical-chemical properties of pedological interest (e.g., dielectric properties of the soil). Additionally, transmission line techniques need large samples to enable measurements at low frequencies, and the use of an open-ended coaxial probe [16] is not recommended either since it generally provides a small sensing volume in the vicinity of the soil probe contact surface, ignoring phenomena that occur deeper. On such a basis, recently, there has been an increasing demand for alternative monitoring techniques that could provide fast measurements using low-cost technology with in situ controls. In this regard, microwave reflectometry [17] is a promising solution, capable of relating the complex dielectric permittivity ɛ^*^ [18] of a material under test (MUT) with some specific characteristics, such as moisture content or the presence of contaminants in the soil. In fact, when the soil is dry or in the presence of pollutants or oil leakages, a detectable change in the dielectric characteristics of the soil occurs. Therefore, considering the above-mentioned limitations, in this work we propose the use of a microwave reflectometry system utilizing a bifilar probe [19,20] and a miniaturized Vector Network Analyzer (m-VNA) [21] to measure the reflection coefficient and then retrieve the dielectric parameters of interest using post-processing. The use of a m-VNA significantly reduces the system cost whilst still ensuring good measurement accuracy. As per the adopted two-rod probe, it provides an easy calibration process, facilitates probe insertion in the soil, and ensures good contact between the probe and soil, even in the case of rocky or crushed stone soils. In addition, this probe configuration bounds the EM field to be in the spacing between the rods, thus increasing the sensing volume, which allows for a comprehensive assessment of the soil under test.

The main objective of this work was to retrieve the dielectric parameters of soils in terms of Debye parameters through a combined approach, making use of traditional FD measurements, SOL (open, short, and load) calibration, full wave modelling and simulations using CST and a minimization routine. The latter is based on minimizing the differences between FD measurements and full wave simulation using a specific probe-model. Firstly, the full wave model was developed and validated against FD measurements. The Debye law better describes the dielectric behavior of materials when compared to the evaluation of the apparent dielectric permittivity [22]. Initially, the proposed method was validated with measurements on reference liquids and then followed by measurements on sand with different water content and sand with different concentrations of oil diesel as a contaminant. The experimental results confirm that using the proposed procedure, different moisture levels in the soil can be monitored and different types of contaminants can be identified.

This paper is organized as follows, Section 3 describes the operating principles of the proposed system, followed by Section 4, which presents the methodological procedure and a thorough description of the experiments. Section 5 summarizes the experimental results and, finally, in Section 6, conclusions and future work are outlined.

## 3. Background

Microwave reflectometry (MR) is a powerful tool employed in several fields of interest [23], ranging from biomedical applications [24,25,26], structural health monitoring [27,28,29,30], characterization of devices [31], soil moisture content monitoring [32,33,34,35] or for leak localization in underground pipes [36,37,38,39]. Typically, as reported in Figure 1, a microwave reflectometry-based monitoring system consists of three major components [40,41]:a probe;the instrument for generating/receiving the electromagnetic (EM) signal;an elaboration unit to acquire and process the measured data.

An EM stimulus propagates along the probe and is partially reflected at the probe-material interface due to permittivity variations. The dielectric characteristics of MUT can be retrieved by analyzing the reflected signal. The reflected stimulus can be measured either in the time domain (time domain reflectometry—TDR) or in the frequency domain (frequency domain reflectometry—FDR). Generally, instruments operating directly in FD, such as VNAs, have the advantage of higher measurement accuracy [42] as calibration procedures can be implemented to minimize systematic errors. However, benchtop VNAs are more expensive than instruments operating directly in the time domain (TD). Recently, low-cost portable VNAs have been made commercially available and they are suitable for measurements over a small range of frequencies (up to 3 GHz). In this paper, reflection measurements in the frequency domain are performed using a miniaturized low-cost VNA, and the reflected signal is then analyzed in terms of the frequency-dependent reflection scattering parameter S_11_(f).

The frequency-dependent dielectric properties of the MUT can be described using a Debye model [43,44], which is a particular case of the Cole-Cole model, which is a very well-referenced model in the literature and can be used to describe the material dispersion in a wide frequency range. The equation for the Cole-Cole model is:(1)$$\varepsilon^{*}=\varepsilon_{\infty }+\frac{\varepsilon_{s}-\varepsilon_{\infty }}{1+{\left(j2\pi f\tau \right)}^{1-\alpha }}$$
where $\varepsilon_{s}$, $\varepsilon_{\infty }$, and τ are three real positive parameters representing, respectively, the relative permittivity at low frequency, the relative permittivity in the limit for high frequency values, and the relaxation time, and *f* denotes the frequency. The α parameter allows the description of different spectral shapes, and in the Debye model (adopted here) it is equal to 0. As is well known, in a Cole-Cole model, α takes a value between 0 and 1, but in this model, it is considered inefficient for numerical analysis [45]. For this reason, for the purposes of the present work, a Debye model was used since for soil and sand the reported values of the α parameter are of the order of ≈0.01 [46]. This choice of model allowed for fast numerical computations, ensuring good accuracy in the working frequency range (up to 3 GHz).

## 4. Materials and Methods

### 4.1. Experimental Setup

In this work, a customized bifilar probe with a rod length of 100 mm made of brass has been adopted. Brass has good corrosion resistance, and it is typically less expensive than other materials such as copper or bronze. It is a strong and durable metal, which prevents the deformation of bars when immersed in granular soil. For the sake of completeness, a CST simulation comparing the conductive performance of the probe in air with the rods made of brass and copper was carried out. As a result, from Figure 2, it can be noted that S_11_ magnitude and phase curves are almost superimposed, demonstrating that small variations between the two materials, especially in terms of electrical conductivity, do not significantly affect the frequency behavior of the probe.

For the specific purposes of this application, the main requirements to be simultaneously met in the probe design phase were simplicity of construction, easy calibration process, and simple insertion into the soil. In addition, this type of probe is easily replicable, cheap, and customizable as it is made simply using two rods, as illustrated in Figure 3. The head of the probe is connected on one side to the two rods through two banana connectors and the other side to a coaxial/wire BNC transition to connect to the m-VNA. Figure 3 presents the model generated in CST Microwave Studio, which was then used to conduct simulations. A discrete port of 50 Ω was embedded in the head of the probe. Figure 3b depicts the actual probe’s configuration, including all geometric considerations.

The complete experimental setup is shown in Figure 4 and consists of a m-VNA, developed by HCXQS (commercially available under the name of nanoVNA) and two rods as described above. The size of the m-VNA is 15 cm × 10 cm × 6 cm. It is low cost and operates from 50 kHz up to 3 GHz. To obtain accurate results, a SOL calibration (short, open, and load) was also performed. The whole experimental setup is shown in Figure 4.

### 4.2. Methodological Procedure and Description of the Experiments

In this work, a methodological procedure has been adopted to accurately retrieve the Debye parameters of soil. This consisted of three main steps: accurate modelling of the two rods; validation of the model using reference liquids; and final testing on soil samples. The subsequent main steps of the procedure are schematized in Figure 5 and can be summarized as follows:(1)Initially, the probe model was optimized through a parametric study using the commercial software CST Microwave Studio. During this optimization procedure, different probe settings such as the dielectric permittivity of the probe head, the discrete port position, and the electrical conductivity of the bars were determined. The optimization procedure was based on the minimum difference between the measured S_11_(f) and that obtained via simulations. This was done so that the developed model, utilizing the optimal probe settings, is a good representation of the experimental setup used in the laboratory;(2)Subsequently, a validation procedure was performed utilizing well-referenced materials (i.e., methanol and isopropyl alcohol, also called prop-2-ol). An experimental campaign to obtain a set of S_11_(f) for different liquids was conducted, each time measuring the temperature of the sample using a thermometer with a tip immersed in the sample. Then the Debye parameters of the MUT at the measured temperature were taken from the literature [47] and loaded into CST. The S_11_(f) as obtained from simulation and measurements were compared;(3)Good agreement between measurements on reference liquids and simulations carried out using the probe model settings as identified in step (1) was achieved, demonstrating the correct modelling of the probe in CST;(4)Finally, the two rods were immersed in the MUT and the unknown Debye parameters were retrieved using an optimization procedure based on the minimization of the differences between the measured S_11_(f) and the modelled S_11,MOD_(f).

The Debye parameter retrieval uses the CST Trust Region Framework optimizer in CST and looks for the three Debye parameters that best match the experimental data, so deducing the dispersion law. This algorithm creates a local linear model around the starting point and defines an initial trust region radius, an area in which probably the model is good. Subsequently, the simulation procedure is repeated until the error between the measurement and the simulation is considered minimal.

As for the measurement campaign, S_11_(f) measurements were carried out with the probe immersed in:Reference materials: air, methanol, and prop-2-ol;Sand with different moisture contents: 0%, 5%, 10%, 15%, 20%, 25%, and 30%;Contaminated sand at different diesel oil percentages: 0%, 5%, 7.5%, and 10%.

The first set of experiments were carried out to establish the correct modeling parameters by assessing the correspondence between the measured S_11_(f) and modeled S_11,MOD_(f). Following that, the second and third sets of experiments consisted of analyzing the probe response in MUT with unknown dielectric properties. In more detail, set #2 was carried out with different moisture levels of the soil and, finally, experimental session #3 was conducted using contaminated sand which had previously been oven dried, so that no spurious moisture could affect the results.

## 5. Experimental Results

### 5.1. Preliminary Experimental Validation

To test the performance of the m-VNA, first a preliminary comparative analysis with a reference benchtop, an accurate but expensive VNA, namely the VNA R&S ZLV6, was conducted. The two rods were left in the air and the S_11_ as a function of frequency was measured using both VNAs. The measured data is presented in Figure 6, and the root mean square error (RMSE) was 0.036.

Considering the good agreement between the measurements in air performed with the m-VNA and with the VNA R&S, the subsequent measurements were carried out, using the m-VNA.

For the probe model validation, measurements in reference materials were carried out and full wave simulations were performed using the material’s well-known dielectric parameters obtained from the NPL report [47] at 20 °C. These parameters are reported in Table 1, for ease of reference.

For the sake of brevity, only the RMSE values of the magnitude of the (measured and simulated) scattering parameter for each reference material are reported. In particular, the RMSE value between S_11_(f) and S_11,MOD_(f) was taken as a figure of merit for estimating the efficiency of the adopted model in properly describing the actual probe; the obtained values are reported in Table 2 and show a good overall agreement between model and measurements for all the considered reference materials.

### 5.2. Experimental Results on Sand with Different Moisture Contents

After the preliminary validation, an experimental campaign was conducted involving sand at progressively higher values of moisture content, starting from 5% and going up to 30% water in the sand in 5% steps.

The measured S_11_ as a function of frequency for sand with different percentages of water are presented in Figure 7, illustrating that when hydration increases, the resonant peaks of |S_11_(f)| shift toward lower frequencies.

In order to retrieve the dielectric characteristics through the CST software, the minimization step and a set of ‘‘initial guesses” for the Debye parameters of dry sand were considered ($\;\varepsilon_{s}$ = 2.52, $\varepsilon_{\infty }$ = 2.47, and τ = 21.5 ps [48]). The best agreement between measured data and simulations for dry sand and dry sand with different moisture content (10%, 20%, and 30%) are reported in Figure 8a–d.

A similar trend to that observed in Figure 8 was obtained for the phase response of S_11_, however these were not included in this paper. All the extracted Debye parameters are reported in Table 3.

These results indicated that $\varepsilon_{s}\;$and $\varepsilon_{\infty }\;$were the parameters that exhibited significant variations for different MUT. In particular, they increase monotonically when increasing the moisture percentage, as can be seen in Figure 9a,b for $\varepsilon_{s}\;$and $\varepsilon_{\infty }$, respectively. This is attributed to the fact that water has a high dielectric constant ($\varepsilon_{r}$ ≈ 80), so it modifies the dielectric parameters of the MUT by strongly affecting the propagation of electromagnetic waves, resulting in a significant increase in both $\varepsilon_{s}$ and $\varepsilon_{\infty }$.

### 5.3. Experimental Results on Contaminated Sand with Diesel Oil

The same kind of minimization procedure implemented for the sand progressively filled with water was carried out for sand contaminated with diesel oil at different percentages, starting from dry sand and then adding 5%, 7.5%, and 10% diesel oil. Figure 10 shows the |S_11_(f)| of the different samples, and it can be noted that as the diesel oil concentration increases, there is a shift of the resonant frequency towards lower values.

The results presented in Figure 11a–d were obtained using the CST software minimization procedure. In addition, in Table 4, the optimized dispersion parameters are reported.

From Table 4, it can be noted that $\varepsilon_{s}$ can be considered as an indicator of the quality of the soil, as reported in the plot shown in Figure 12.

## 6. Conclusions and Future Works

In this paper, a specific procedure for monitoring the moisture content and the presence of diesel oil contaminants in the soil was investigated. The proposed approach relies on the evaluation of the Debye parameters, and thus the dielectric characteristics of the soils. In particular, the adopted procedure involves a minimization routine between FDR measurements carried out with a low-cost, but accurate m-VNA and numerical data obtained through simulations in CST Microwave Studio. The probe model developed in CST was validated using reference liquids with well-known dielectric properties. The results show that the different water or contaminant content can be discriminated against by considering the Debye dielectric parameters. These three dispersion parameters allow for much more useful and complete information for monitoring purposes than comparing only the apparent dielectric permittivity. In addition, despite the need to perform a specific routine after the measurements, this procedure is very quick and takes a few minutes, making the proposed method quasi real-time. Finally, it is important to note that, unlike other expensive instruments, the proposed method is implemented with low cost and good measurement accuracy. Further work will be dedicated to the development of customized software, overcoming the limited use of CST Microwave studio software and related license costs. In addition, the proposed procedure will be further improved through the development of a specific calibration procedure relating the output data in terms of dielectric parameters to the measured temperature of the MUT. For this purpose, some measurements will be carried out in a climatic chamber on soil samples at a specific temperature, identifying the dielectric parameters in standard conditions at varying temperatures. However, these results, although preliminary, demonstrate the feasibility of employing a portable and low-cost VNA with a customized bifilar probe to sense the dielectric variation of soils, monitoring hydration state or detecting possible contaminant agents.

## Figures and Tables

**Figure 1 sensors-22-07805-f001:**
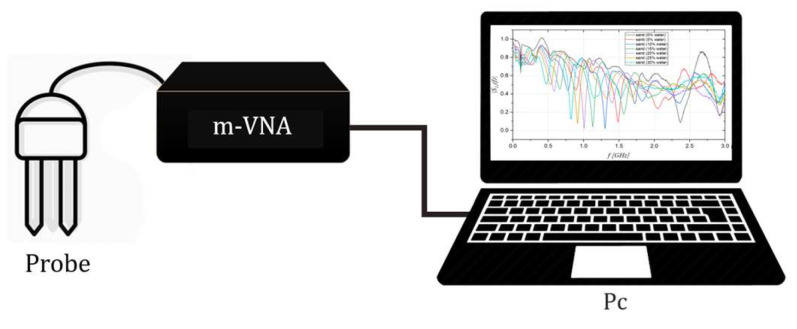
A schematic of a microwave reflectometry-based measuring system.

**Figure 2 sensors-22-07805-f002:**
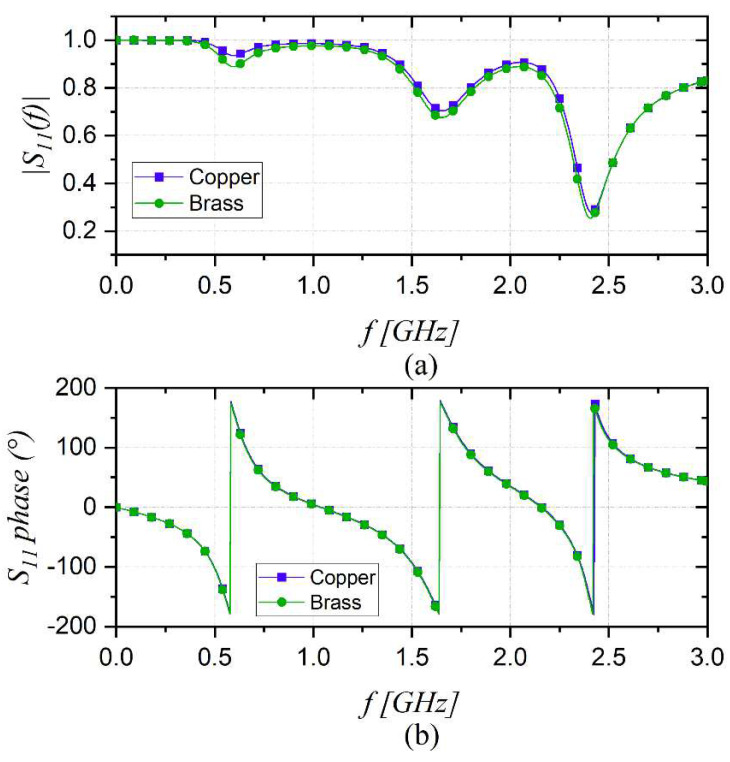
CST simulations comparing the performance of the probe in air with the rods made of brass and copper. (**a**) S11 magnitude; (**b**) S11 phase.

**Figure 3 sensors-22-07805-f003:**
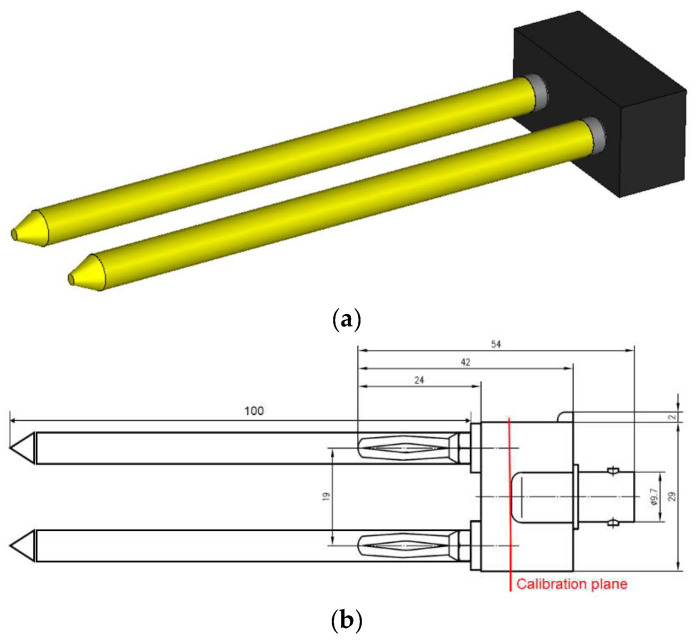
Configuration of the modelled coaxial probe. (**a**) CST-created 3-D model view; (**b**) Schematic of the probe.

**Figure 4 sensors-22-07805-f004:**
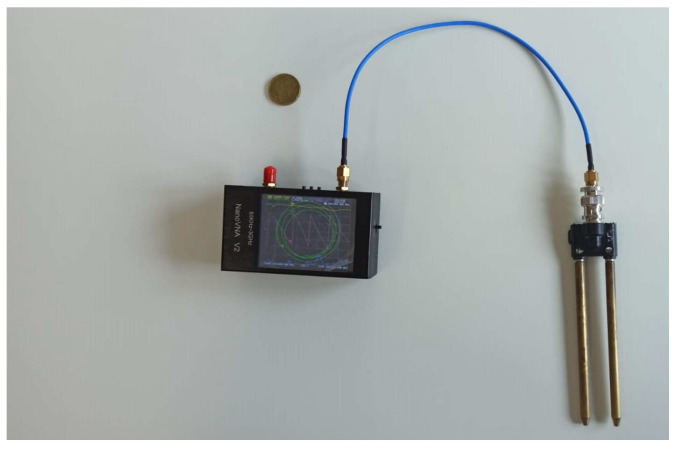
The experimental setup.

**Figure 5 sensors-22-07805-f005:**
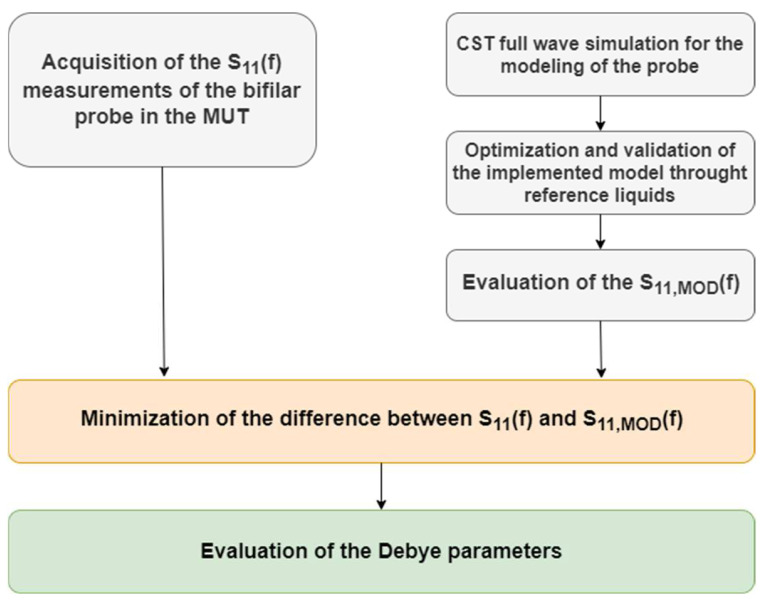
Schematic of the proposed procedure to retrieve the MUT’s Debye parameters, combining both numerical simulations (CST) and experimental data.

**Figure 6 sensors-22-07805-f006:**
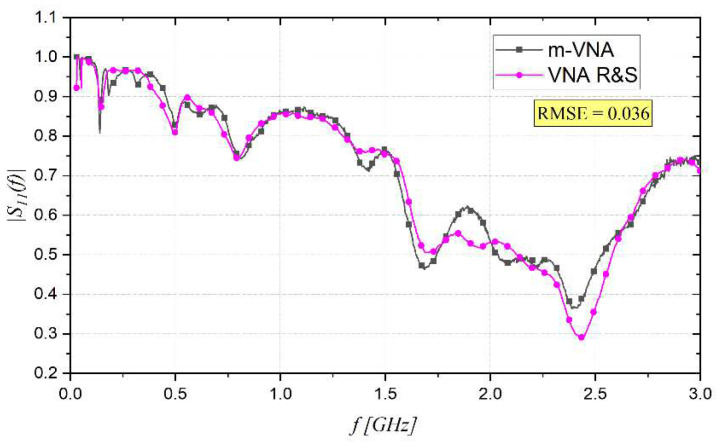
Measurement of the S_11_ as a function of frequency using m-VNA and VNA R&S in air.

**Figure 7 sensors-22-07805-f007:**
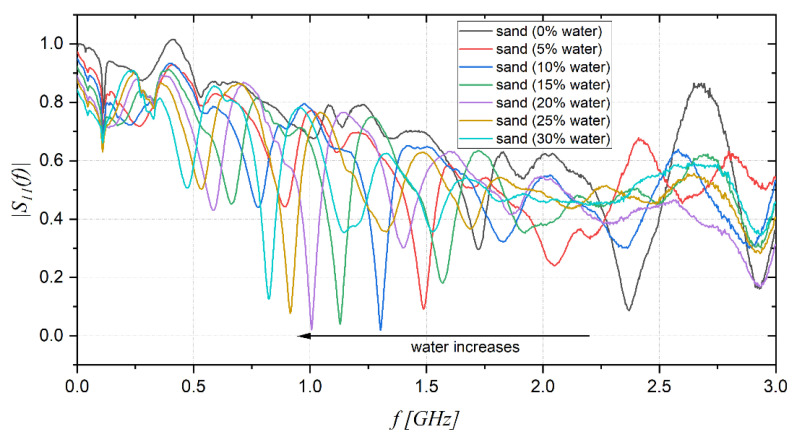
|S_11_(f)| measurements of sand with different moisture content.

**Figure 8 sensors-22-07805-f008:**
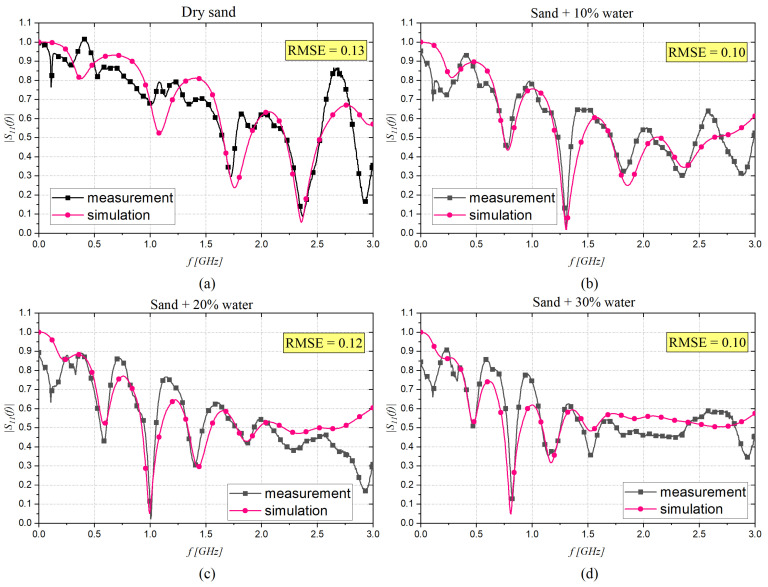
Magnitude of the S11 (f) measured with the probe immersed in sand with different concentrations of water: (**a**) 0%, (**b**) 10%, (**c**) 20%, and (**d**) 30%.

**Figure 9 sensors-22-07805-f009:**
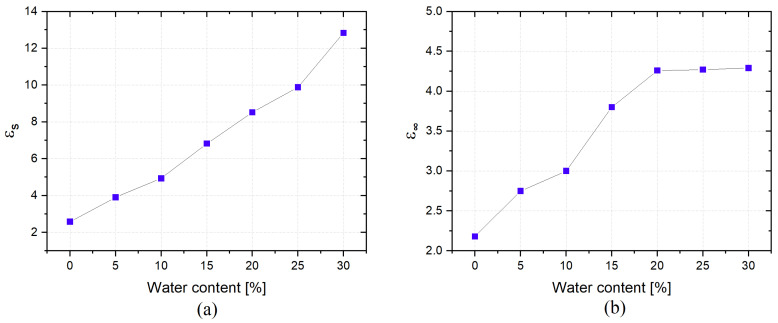
(**a**) $\varepsilon_{s}$ and (**b**) $\varepsilon_{\infty }$ at different soil moisture levels.

**Figure 10 sensors-22-07805-f010:**
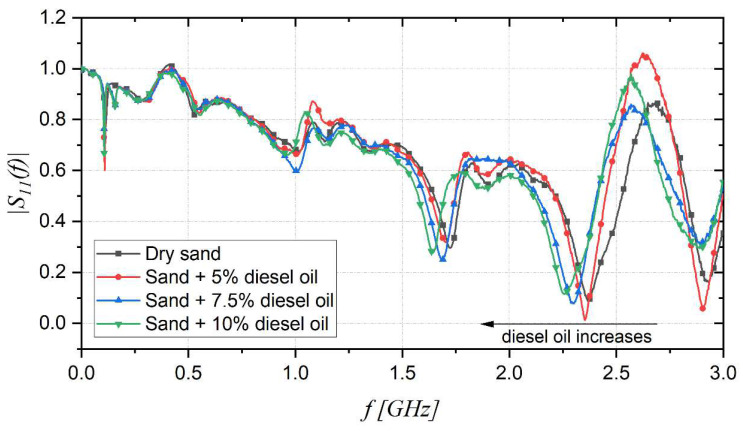
Measured |S_11_(f)| with the probe immersed in sand containing varying concentrations of diesel oil.

**Figure 11 sensors-22-07805-f011:**
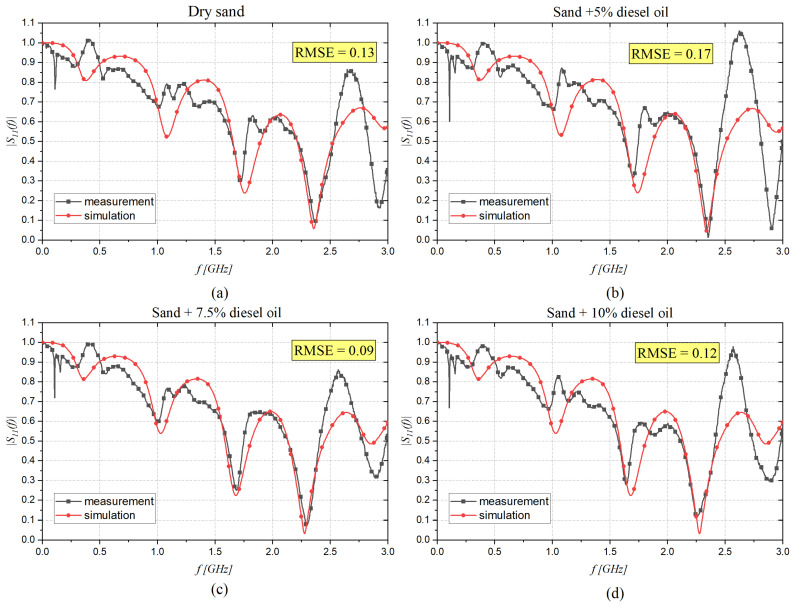
Magnitude of the S_11_ (f) measured with the probe immersed in sand with different concentrations of diesel oil: (**a**) 0%, (**b**) 5%, (**c**) 7.5%, and (**d**) 10%.

**Figure 12 sensors-22-07805-f012:**
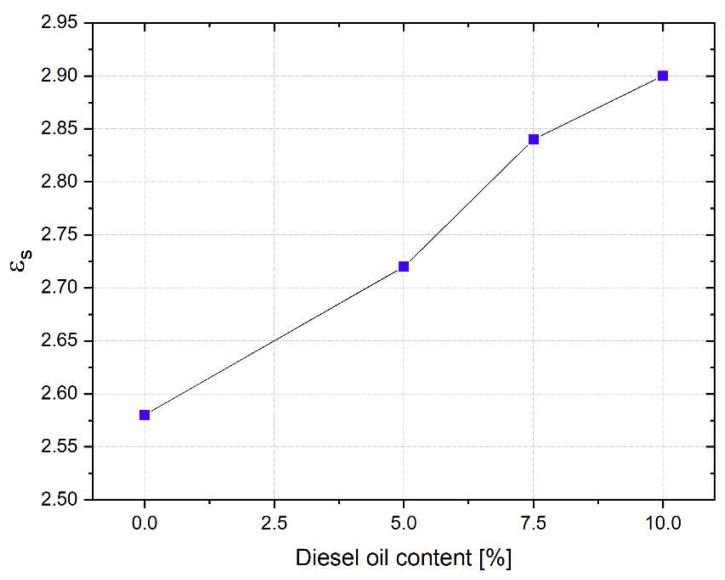
Trend of $\varepsilon_{s}$ at different concentrations of diesel oil in soil.

**Table 1 sensors-22-07805-t001:** Dielectric parameters of a Debye model for the reference liquids from [36].

MUT	$\varepsilon_{s}$	$\varepsilon_{\infty }$	τ (ps)
Methanol	33.64	5.65	56.39
Prop-2-ol	20.11	3.56	453.43

**Table 2 sensors-22-07805-t002:** Displays the RMSE values obtained when comparing experimental data and simulations with the probe in air, methanol, and prop-2-ol.

MUT	RMSE
Air	0.10
Methanol	0.09
Prop-2-ol	0.06

**Table 3 sensors-22-07805-t003:** Extracted dielectric parameters of the Debye dispersion model for sand with different moisture contents.

MUT	$\varepsilon_{s}$	$\varepsilon_{\infty }$	τ (ps)
Sand (dry)	2.58	2.18	21.50
Sand (water = 5%)	3.90	2.75	23.20
Sand (water = 10%)	4.93	3.00	25.00
Sand (water = 15%)	6.82	3.80	24.90
Sand (water = 20%)	8.52	4.26	23.00
Sand (water = 25%)	9.88	4.27	23.50
Sand (water = 30%)	12.83	4.29	24.00

**Table 4 sensors-22-07805-t004:** Extracted dielectric parameters of the Debye dispersion model for sand with different diesel oil contents.

MUT	$\varepsilon_{s}$	$\varepsilon_{\infty }$	τ (ps)
Sand (dry)	2.58	2.18	21.50
Sand (diesel oil = 5%)	2.72	2.23	21.50
Sand (diesel oil = 7.5%)	2.84	2.29	21.50
Sand (diesel oil = 10%)	2.90	2.21	21.50

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
