# Peer review of "A Method for Extracting Debye Parameters as a Tool for Monitoring Watered and Contaminated Soils"

_sensors, 2022, doi:10.3390/s22207805_

Round 1

Reviewer 1 Report

The authors present a method for monitoring contaminated soils. The work deals with the detection of moisture and the presence of oil in sand. The work is of interest to environmental control.

However, the manuscript (and research) needs some improvements, in my opinion:

1. The use of Debye instead of a more "robust" model like Cole-Cole; since,  as presented widely in the literature, it is a better descriptor of this kind of complex media.

2. A description of temperature control lacks in the research. This is a critical issue when measuring the dielectric response of materials with water content.

Author Response

We thank the Reviewer for appreciating the work. Please check the attachment.

Author Response

(The authors gave the same response as above.)

Round 2

Reviewer 2 Report

The authors have satisfactorily addressed all my prior comments, and the manuscript now shows quality enough for consideration. I have only one last suggestion:

Considering the justification of the selection of brass as conductive material, in addition to the economic convenience the authors are suggested to include a brief discussion on the expected performance of the probe with other more conductive materials, such as copper.
